# Attention-Gated Networks
# for Improving Ultrasound Scan Plane Detection

**Jo Schlemper**[1], **Ozan Oktay**[1], **Liang Chen**[1], **Jacqueline Matthew**[2],
**Caroline Knight**[2], **Bernhard Kainz**[1], **Ben Glocker**[1], and **Daniel Rueckert**[1]

[1]Biomedical Image Analysis Group, Imperial College London, London, UK
[2]King's College London, London, UK
jo.schlemper11@imperial.ac.uk

## Abstract

In this work, we apply an attention-gated network to real-time automated scan plane detection for fetal ultrasound screening. Scan plane detection in fetal ultrasound is a challenging problem due the poor image quality resulting in low interpretability for both clinicians and automated algorithms. To solve this, we propose incorporating *self-gated soft-attention mechanisms*. A soft-attention mechanism generates a gating signal that is end-to-end trainable, which allows the network to contextualise local information useful for prediction. The proposed attention mechanism is generic and it can be easily incorporated into any existing classification architectures, while only requiring a few additional parameters. We show that, when the base network has a high capacity, the incorporated attention mechanism can provide efficient object localisation while improving the overall performance. When the base network has a low capacity, the method greatly outperforms the baseline approach and significantly reduces false positives. Lastly, the generated attention maps allow us to understand the model's reasoning process, which can also be used for weakly supervised object localisation.

## 1 Introduction

Fetal ultrasound screening is an important diagnostic protocol to detect abnormal fetal development. Abnormal development is one of the leading causes for perinatal mortality world-wide. During screening examination, multiple anatomically standardised [15] scan planes are used to obtain biometric measurements as well as identifying abnormalities such as lesions. While 2D ultrasound is the preferred approach for examination due to its low cost and real-time capabilities, ultrasound suffers from low signal-to-noise ratio and image artefacts. As such, diagnostic accuracy and reproducibility is limited and requires a high level of expert knowledge and training. Therefore, automated scan plane detection algorithms can help training experts, facilitate non-expert examination, support consistent data acquisition and make diagnostics more robust.

Automated scan plane detection poses many challenges: Firstly, during the examination, the majority of time is spent exploring the present anatomy. As such, there are a large number of background labels and a significant class imbalance must be considered. Secondly, even if the object of interest is localised, it may not have reached the ideal scanning plane for diagnosis and hence the frame may be labelled as background; Therefore, in addition to understanding the global context, it is essential to understand the small differences in local structures to detect a correct plane. In the past, several approaches were proposed [28, 3], however, they are computationally expensive and cannot be deployed for the real-time application.

1st Conference on Medical Imaging with Deep Learning (MIDL 2018), Amsterdam, The Netherlands.

In recent years, deep learning and convolutional neural networks (CNNs) have become popular approaches for a variety of medical image classification problems, including classification of Alzheimer's disease [20], lung nodule in CT/X-ray [34], skin lesion [4, 7], anatomy [19] and the views for echo-cardiograms [13].An extensive list of applications can be found in [9, 29]. In [2] the authors propose a CNN architecture called *Sononet* to solve the standard plane classification problem during fetal ultrasound examination. The proposed approach achieves very good performance in real-time plane detection, retrospective frame retrieval (retrieving the most relevant frame) and weakly supervised object localisation. However, despite its success, the method suffers from relatively low precision and especially struggles differentiating anatomically related cardiac views. We argue that the reason for this is that Sononet is good at aggregating global information but it cannot preserve local information well. Moreover, the heuristics employed for the object localisation requires guided backpropagation, which limits the object localisation speed that can be achieved.

In fact, we claim that the inability to exploit local information is a common problem in medical image analysis: in many of these scenarios, typically, the object of interest is very small (e.g. lesions, local deformity, etc.) compared to the size of the input image, which can be high resolution 2D, 3D or 4D data. Such situation requires to tackle the object detection and classification problem as a two-stage process. In this work, we introduce *soft-attention* in the context of medical image classification. Attention is a modular mechanism that allows to efficiently exploit localised information, which also provides soft object localisation during forward pass. In this work, we demonstrate the usefulness of such attention mechanism by applying the proposed approach to improve the scan plane detection for fetal ultrasound screening.

## 1.1 Related Work

Attention mechanisms were first popularised in the context of natural language processing [21], such as machine translation [1, 12]. In these settings often recurrent neural networks are employed to model a sequence of text. In particular, given a sequence of text and a current word, a task is to extract a next word in a sentence generation or translation. The idea of attention mechanisms is to generate a *context* vector which assigns weights on the input sequence. Thus, the signal highlights the salient feature of the sequence conditioned on the current word while suppressing the irrelevant counter-parts, making the prediction more contextualised. Attention mechanisms can further be separated into two types: soft-attention and hard-attention. In soft-attention, continuous functions (e.g. soft-max) are used to assign the attention weight on the input, making it fully differentiable. In comparison, hard-attention models propose specific words by sampling from the weights. As the sampling operation is not differentiable, hard-attention is trained using the gradient of the likelihood term generated by Monte-Carlo sampling [26].

In computer vision, attention mechanisms are applied to a variety of problems, including image classification [6, 23, 32], segmentation [18], action recognition [10, 16, 24], image captioning [26, 11], and visual question answering [27, 14]. In the context of medical image analysis, attention models have been exploited for medical report generation [31, 30] as well as joint image and text classification [25]. However, for standard medical image classification, despite the importance of local information, only a handful of works use attention mechanisms [17, 5]. In these methods, either bounding box labels are available to guide the attention, or the local context is extracted by a hard-attention model (i.e. region proposal followed by hard-cropping). In our work, we propose incorporating self-gating, a soft-attention approach that is end-to-end trainable. This also does not require any bounding box labels and backpropagation-based saliency map generation as in [2].

## 1.2 Contributions

- We introduce a self-gated, soft-attention mechanism in the context of pure medical image classification. We apply the proposed model to real-time fetal ultrasound scan plane detection and show its superior classification performance over the baseline approach.

- We demonstrate that the proposed attention mechanism provides fine-scale attention maps that can be visualised, with minimal computational overhead, which is a crucial step towards explainable deep learning.

- Finally, we show that attention maps can used for fast (weakly-supervised) object localisation, demonstrating that the attended features indeed correlates to the anatomy of interest.

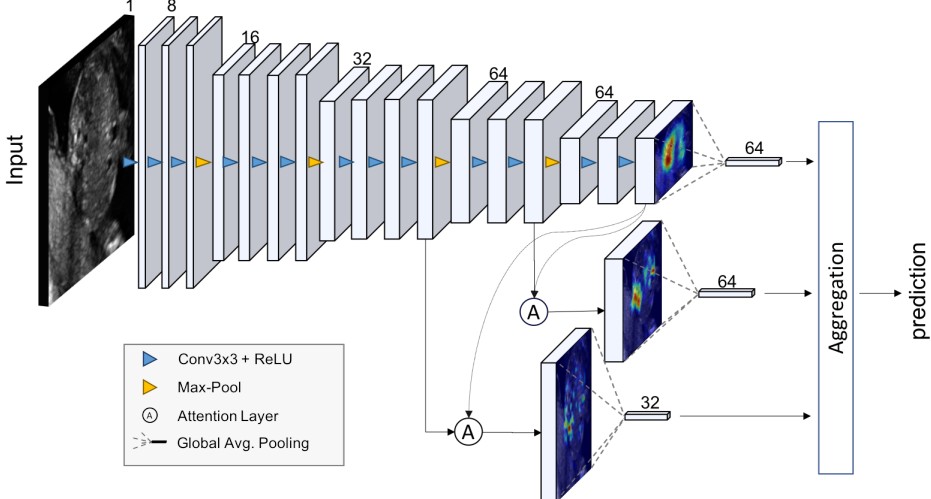

Figure 1: The schematics of the proposed network, termed *Attention-Gated Sononet*. The proposed attention units are incorporated in layer 11 and layer 14.

## 2  Methodology

**Sononet:** We will first review *Sononet* [2], which will be the baseline architecture for our discussion. Sononet is a CNN architecture with two components: a feature extractor module and an adaptation module. In the feature extractor module, the first 17 layers (counting max-pooling) of the VGG network [22] is used to extract discriminant features. Note that the number of filters are doubled after each of the first three max-pooling operations. In the adaptation module, the number of channels are first reduced to the number of target classes $C$. Subsequently, the spatial information is flattened via channel-wise global average pooling. Finally, a soft-max operation is applied to the resulting vector and the entry with maximum activation is selected as the prediction. As the network is constrained to classify based on the reduced vector, the network is forced to extract the most salient features for each class. Owing to this, Sononet obtains well-localised feature maps before the pooling layer, which can also used to perform weakly-supervised localisation (WSL) with high accuracy. However, while the global average pooling is attractive as it can quickly aggregate the spatial context, it does not have the capacity to preserve local information. As such, if two frames have very similar global appearance, it cannot well distinguish them. In the case of scan plane detection, this is manifested as it results in a low accuracy for multiple cardiac views, where each view contains similar underlying anatomy but only differ by the plane orientation.

**Attention Unit for Medical Image Analysis:** Our work is inspired by [6], where the authors have introduced a similar attention mechanism for a classification problem in the computer vision domain. Let $\mathcal{F}^s = \{\mathbf{f}_i^s\}_{i=1}^n$ be the activation map of a chosen layer $s \in \{1, \ldots, S\}$, where each $\mathbf{f}_i^s$ represents the pixel-wise feature vector of length $C_s$ (i.e. the number of channels). Let $\mathbf{g} \in \mathbb{R}^{C_g}$ be a global feature vector extracted just before the final soft-max layer of a standard CNN classifier. In this case $\mathbf{g}$ encodes global, discriminative, relevant information about the objects of interest. The idea is to consider each $\mathbf{f}_i^s$ and $\mathbf{g}$ jointly to attend the features at each scale $s$ that is relevant to the coarse scale features (i.e. object-ness) represented by $\mathbf{g}$. To this end, the notion of a *compatibility score* $\mathcal{C}(\mathcal{F}^s, \mathbf{g}) = \{c_i^s\}_{i=1}^n$ is defined and is given by an *additive* attention model:

$$c_i^s = \langle \mathbf{\Psi}, \mathbf{f}_i^s + \mathbf{g} \rangle, \tag{1}$$

where $\langle \cdot, \cdot \rangle$ is the dot product and $\mathbf{\Psi} \in \mathbb{R}^{C_s}$ is a learnable parameter. In the case where $\mathbf{f}_i^s$ and $\mathbf{g}$ have different dimensions, a learnable weight $\mathbf{W}_g \in \mathbb{R}^{C_s \times C_g}$ is used to match the dimensionality of $\mathbf{g}$ to $\mathbf{f}_i^s$. Once the compatibility scores are computed, they are passed through soft-max operation to obtain the normalised attention coefficient: $\alpha_i^l = e^{c_i^l} / \sum_i e^{c_i^l}$. Finally, at each scale $s$, a weighted sum $\mathbf{g}^s = \sum_{i=1}^n \alpha_i^s \mathbf{f}_i^s$ is computed, and the final prediction is given by fitting a fully connected layer on $\{\mathbf{g}^1 \ldots \mathbf{g}^S\}$. By constraining the prediction to be done from the weighted sum, the network is forced to learn the most salient features that contribute to the class. Therefore the attention coefficients

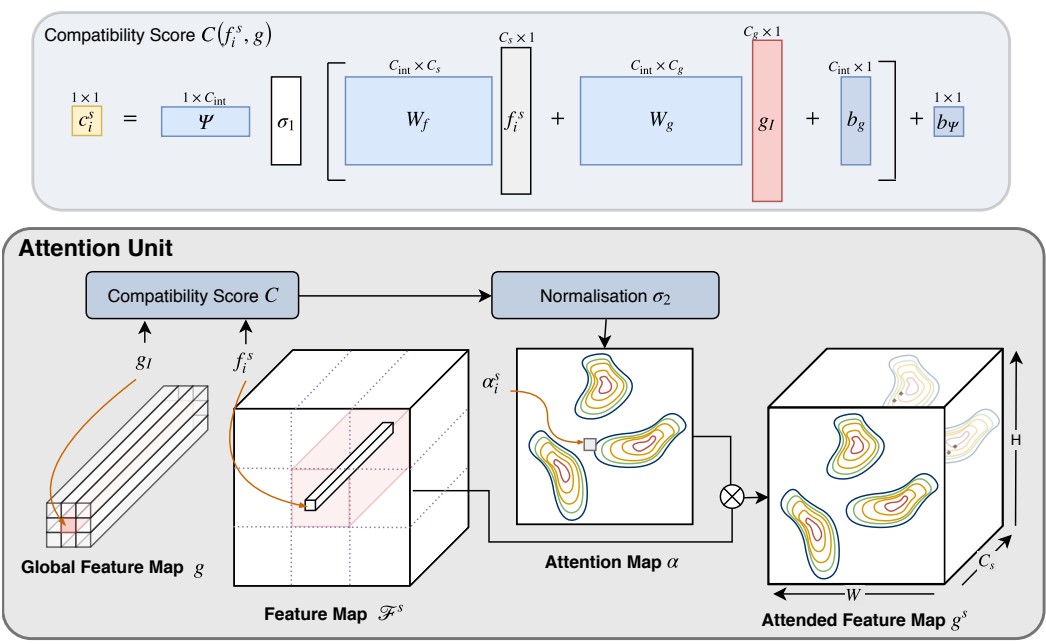

Figure 2: The proposed grid attention block. The global gate signal $\mathbf{g}_I$ is shared for the region indicated in red. Tensor dimensions for the compatibility score computation are shown.

$\{\alpha_i^l\}$ identify salient image regions, amplify their influence, and suppress irrelevant information in background regions. In this work, we consider a more general attention mechanism:

$$c_i^s = \mathbf{\Psi}\sigma_1\left(\mathbf{W}_f\mathbf{f}_i^s + \mathbf{W}_g\mathbf{g} + \mathbf{b}_g\right) + \mathbf{b}_\psi, \tag{2}$$

where a gating unit is characterised by a set of parameters $\Theta_{att}$ containing: linear transformations $\mathbf{W}_f \in \mathbb{R}^{C_{int} \times C}$, $\mathbf{W}_g \in \mathbb{R}^{C_{int} \times C_g}$, $\mathbf{\Psi} \in \mathbb{R}^{C_{int}}$ and bias terms $\mathbf{b}_\psi \in \mathbb{R}$, $\mathbf{b}_g \in \mathbb{R}^{C_{int}}$. $\sigma_1$ is a nonlinearity chosen to be ReLU: $\sigma_1(\mathbf{x}) = \max(0, \mathbf{x})$. By introducing attention mechanism, the features $\mathcal{F}^s$ branches out to two paths: one to extract the global feature vector and one which passes through gating for the prediction. The benefit of the generalised approach is that firstly, we speculate that introducing $\mathbf{W}_f$ allows the fine-scale layer to focus less on generating a signal that is compatible to $\mathbf{g}$, which helps it focus on learning the discriminant features. Secondly, by introducing $\mathbf{W}_f$, $\mathbf{W}_g$ and $\sigma_1$, we allow the network to learn nonlinear relationships between the vectors, which is more expressive. This is important because in medical imaging, images are inherently noisy and the region of interest is often highly non-homogeneous. Therefore, a linear compatibility function may be too sensitive to such fluctuation. Note that $\{\mathbf{W}_f\}$ and $\{\mathbf{\Psi}, \mathbf{b}_\psi\}$ are implemented as $1 \times 1$ convolution layer and $\{\mathbf{W}_g, \mathbf{b}_g\}$ is a fully connected layer.

Similarly, for normalising the compatibility coefficients, we also argue that soft-max operation may not be the optimal approach as it typically produces sparse output. This can be over-sensitive to local changes even though we want the network to attend to the region with high variability. An alternative is a sigmoid unit. However, we observe that a sigmoid often suffers from gradient saturation problems. As such, the best working strategy is to first subtract the $\min_i\{\alpha_i^s\}_{i=1}^n$ from all attention coefficients to align the minimum value to be 0. Then, we divide each element by $\sum_i \alpha_i^s$. This is similar to soft-max but does not sparsify the attention map.

**Grid Attention:** As seen above, the global feature vector $\mathbf{g}$ is a 1D vector incorporating information from all spatial context. In the context of medical imaging, since most objects of interest are extremely localised, flattening may have the disadvantage of losing important spatial context. In fact in many cases, a few max-pooling operations are sufficient to infer the global context without explicitly using the global pooling. Therefore, we propose a *grid attention* mechanism. The idea is to use the feature map just before the global pooling as the gridded instance of $\mathbf{g}$. For example, given an input size of $N_x \times N_y$, after $r$ max pooling operations, the tensor size is reduced to $N_x/(2^r) \times N_y/(2^r)$. To generate the attention map, we upsample the coarse grid to match the spatial resolution of $\mathcal{F}^s$ (see

Figure 2). In this way, the attention mechanism has more flexibility in terms of what to focus on a regional basis. Concretely, in terms of implementation, $\{\mathbf{W}_g, \mathbf{b}_g\}$ is now replaced by a $1 \times 1$ convolution. For upsampling, we chose to use bilinear upsampling. Note that the upsampling can be replaced by a learnable weight, however, we did not opt for this for the sake of simplicity.

**Attention Gates in Sononet:** The proposed attention mechanism is incorporated in the Sononet architecture to better exploit local information. In the modified architecture, termed Attention-Gated Sononet (*AG-Sononet*), we remove the adaptation module. The final layer of the extraction module is used as gridded global feature map $\mathbf{g}$. We apply the proposed attention mechanism to layer 11 and 14 just before pooling, as shown in Figure 1. After the attention map $\{\alpha_i^s\}$'s are obtained, the weighted average over the spatial axes is computed for each channel in the feature map, yielding a vector of length $C^s$ at scale $s$. In addition, we also perform the global average pooling on the coarsest scale representation and use it for the final classification. This is because we hypothesise that the coarsest scale representation is still useful for the classification when fine-scale features are unnecessary.

**Aggregation Strategy:** Given the attended feature vectors at different scales, we combine them for the final prediction. We highlight that the aggregation strategy is flexible and that it can be adjusted depending on the target problem. However, the aggregation strategy also influences how the network learns; simple aggregation might not enforce the network to learn the most useful gating mechanism. The simplest approach is to fit a separate fully connected (FC) layer at each scale, and make separate predictions. The final prediction is then given by either weighted mean or max operations. This approach ensures that the network learns relevant attributes of the classes at each scale, and hence the learning process is more stable. The alternative is to first concatenate the feature vectors and fit one FC layer for the prediction. In theory, this strategy should perform better as it allows the network to combine the information at different scales. However, we observe that this approach is non-trivial to train. Since the network tends to pick up coarse-scale features quickly, it quickly abandons the gating paths for finer scales and gets stuck in a local minimum. We attempted using deep-supervision [8] to force each scale to learn a useful prediction jointly. However in this case, the network again obtains suboptimal performance. We speculate that this is because the network tries to allocate resources for individual-scale prediction and joint scale prediction simultaneously, which are conflicting in nature. The simplest and the most stable approach is to first let the network learn the prediction at each scale. After the network has converged, we fit a new FC layer on top of the predictions at each scale and let the network fine-tune itself for the joint prediction. Thus, the network discards the features that are predicted by other scales and focuses on subtle differences that can only be observed at a given scale. We denote the model which uses simple averaging of individual predictions as *AG-Sononet*, the deep supervision model as *AG-Sononet-DS* and the fine-tuned model as *AG-Sononet-FT*.

## 3  Experiments and Results

In this section, the proposed model is compared against Sononet in terms of classification performance, model capacity, and computation time. In addition, we compare different aggregation strategies discussed above: *AG-Sononet*, *AG-Sononet-DS*, and *AG-Sononet-FT*.

**Evaluation Datasets:** Our dataset consisted of 2694 2D ultrasound examinations of volunteers with gestational ages between 18 and 22 weeks. Image acquisition protocol is specified in [2]. The dataset contains 13 types of standard scan planes and background, complying the standard specified in the UK National Health Service (NHS) fetal anomaly screening programme (FASP) handbook [15]. The standard scan planes are: Brain (Cb.), Brain (Tv.), Profile, Lips, Abdominal, Kidneys, Femur, Spine (Cor.), Spine (Sag.), 4CH, 3VV, RVOT, LVOT. The data was cropped to central $208 \times 272$ to prevent the network from learning the surrounding annotations shown in the ultrasound scan screen. The dataset was split into training (122233), validation (30553) and testing (38243) subsets. For preprocessing, we whitened our data (normalised each image by substracting the mean intensity and divide by the variance). For training, we used the following data augmentation: horizontal and vertical translation of $\pm 4$ pixels, horizontal flips, rotation of $\pm 25 \deg$ and zoom of factor $s \in [0.7, 1.3]$. This generates a dataset at least $40000\times$ bigger than the original.

For evaluation, we used accuracy, precision, recall, F1, the number of parameters and execution speed. Note that due to large class imbalance, it is important to take the macro-averaging for precision, recall and F1: e.g $\text{recall}_{\text{macro}} = (\text{recall}_{c_1} + \cdots + \text{recall}_{c_n})/\{ \text{ the number of classes } \}$. Furthermore, we also

qualitatively study the attention map generated to highlight that the network indeed attends salient local regions.

**Training:** Note that due to the nature of fetal ultrasound screening, the background label dominates the dataset. Due to large class imbalance, the training is not straightforward. In addition, background frames could contain the anatomy of interest, yet it might be classified as background as the plane is not a standard plane. Therefore, an appropriate ratio between all classes and background is important. We used a weighted sampling strategy: the sampling "probability" of an image from class $c$ is given by $1/n_c$, where $n_c$ is the number of images in class $c$. For the background label, we used $13/n_c$, where 13 is the number of the standard scan planes. In this way, we expect to see one background image for every standard scan plane. We used cross entropy loss and the network was optimised using Stochastic Gradient Descent with Nesterov momentum ($\rho = 0.9$). The initial learning rate was set to 0.1, which was subsequently reduced by a factor of 0.1 for every 100 epoch. We also used a warm-start learning rate of 0.01 for the first 5 epochs. Each network was trained for 300 epochs. The batch size was set to 64. $\ell_2$ weight regularisation was used with $\lambda = 10^{-4}$.

**Implementation Details:** We modified the baseline Sononet architecture slightly: instead of using 2 convolution layers for the first 2 feature scales and 3 convolution layers for the last 3 feature scales, we used 3 layers for the first 3 and 2 layers for the last 2 feature scales. The architecture for AG-sononet is shown in 1. As discussed, training AG-sononet is slightly more tricky as the optimal gating mechanism may not be necessarily learnt. However, we observed that the simplest approach to achieve the desired gating mechanism was to initialise AG-Sononet with a partially trained Sononet. We compare our models with different capacities, with initial number of features 8, 16 and 32. Our implementation in PyTorch library is publicly available[1].

**Results:** Table 1 summarises the performance of the models. In general, AG-Sononets improve the results over Sononet at all capacity levels. In particular, AG-Sononets higher precision. AG-Sononets reduces false positive examples because the gating mechanism suppresses background noise and forces the network to make the prediction based on class-specific features. As the capacity of Sononet is increased, the gap between the methods are tightened, but we note that the performance of AG-Sononets is also close to the one of Sononet with double the capacity. In addition, the advantage of AG-Sononets is that it can provide attention maps for no extra computational cost (shown below). Therefore, attention-mechanism allows the network to allocate all resources on the most salient aspect of the problem, and can achieve higher performance with minimal number of parameters. In Table 2, we show the class-wise F1, precision and recall values for AG-Sononet-FT-8. The improvement over Sononet is indicated in brackets. In [2], it was highlighted that the model often confuses between cardiac views as they appear anatomically similar. The situation is notably improved, with statistically significant improvement for 4CH and 3VV ($p < 0.05$) due to fine-scale aggregating differences. However, these views remained challenging. We see that the precision increased by around 5% for kidney, profie and spines, as well as on average 3% for cardiac views. In some cases, we see minor reduction in recall rates. We believe that this is because the network may have become slightly more conservative when predicting the class labels.

**Attention Map Analysis and Object Localisation** In Figure 3, we show the attention map of AG-Sononet. AG-1 and AG-2 are the attention map applied at layer 11 and 14 respectively. AG-3 is the attention map of the final layer (the coarsest). In this case, we do not use attention gates, however, we use activation maps with $C \in \{64, 128, 256\}$ channels depending on the capacity. In order to visualise class-specific attention, we employed Class Activation Mapping (CAM) [33]. AG-all is obtained by taking the mean of the attention maps which are all normalised to have the maximum value 1. Recall that AG-Sononet simply obtains mean of the predictions at each image scale. As such, the attention maps pinpoint the class-specific information at all scales. In Figure 4, we show the attention map of AG-Sononet-FT. In this case, the aggregation layer relearns how to optimally combine the features at different scales. Fine-scale features do not necessarily highlight the whole object of interest, but it highlights key information within it that cannot be observed by the coarse scale representation. Similarly in some cases, fine-scale features seem to not learn anything if the prediction can be done by coarser scales.

Finally, in Figure 5, we show the attention maps of AG-Sononet-FT across different subjects, together the bounding box annotation generated using the attention maps (see Appendix for the heuristics). We see that the network consistently focuses on the object of interest, which indicates that the network

---

[1] https://github.com/ozan-oktay/Attention-Gated-Networks

Table 1: Test results for standard scan plane detection. Number of initial filters is denoted by the postfix "-$n$". Time taken for forward (Fwd) and backward (Bwd) passes were recorded in milliseconds.

| Method | Accuracy | F1 | Precision | Recall | Fwd/Bwd ($ms$) | #parameters |
|---|---|---|---|---|---|---|
| Sononet-8 | 0.969 | 0.899 | 0.878 | 0.922 | 1.36/2.60 | 0.16M |
| AG-Sononet-8 | 0.976 | 0.921 | 0.911 | **0.933** | 1.86/3.46 | 0.18M |
| AG-Sononet-DS-8 | 0.975 | 0.918 | 0.907 | 0.929 | 1.92/3.51 | 0.18M |
| AG-Sononet-FT-8 | **0.977** | **0.922** | **0.916** | 0.929 | 1.92/3.47 | 0.18M |
| Sononet-16 | 0.977 | 0.923 | 0.916 | 0.931 | 1.45/3.92 | 0.65M |
| AG-Sononet-16 | 0.976 | 0.925 | 0.917 | 0.932 | 1.88/5.13 | 0.70M |
| AG-Sononet-DS-16 | **0.978** | 0.924 | 0.919 | 0.929 | 1.90/5.19 | 0.71M |
| AG-Sononet-FT-16 | **0.978** | **0.929** | **0.924** | **0.934** | 1.94/5.13 | 0.70M |
| Sononet-32 | 0.979 | 0.931 | 0.924 | **0.938** | 2.40/6.72 | 2.58M |
| AG-Sononet-32 | **0.980** | 0.932 | 0.928 | 0.937 | 3.01/8.74 | 2.79M |
| AG-Sononet-DS-32 | 0.978 | 0.929 | 0.921 | 0.937 | 2.98/8.81 | 2.80M |
| AG-Sononet-FT-32 | **0.980** | **0.933** | **0.931** | 0.935 | 2.92/8.68 | 2.79M |

Table 2: Class-wise performance for AG-Sononet-FT-8. In bracket shows the improvement over Sononet-8. Bold highlights the improvement more than 0.02.

| | Precision | Recall | F1 |
|---|---|---|---|
| Brain (Cb.) | 0.988 (-0.002) | 0.982 (-0.002) | 0.985 (-0.002) |
| Brain (Tv.) | 0.980 ( 0.003) | 0.990 ( 0.002) | 0.985 ( 0.003) |
| Profile | 0.953 ( **0.055**) | 0.962 ( 0.009) | 0.958 ( **0.033**) |
| Lips | 0.976 ( **0.029**) | 0.956 (-0.003) | 0.966 ( 0.013) |
| Abdominal | 0.963 ( 0.011) | 0.961 ( 0.007) | 0.962 ( 0.009) |
| Kidneys | 0.863 ( **0.054**) | 0.902 ( 0.003) | 0.882 ( **0.030**) |
| Femur | 0.975 ( 0.019) | 0.976 (-0.005) | 0.975 ( 0.007) |
| Spine (Cor.) | 0.935 ( **0.049**) | 0.979 ( 0.000) | 0.957 ( **0.026**) |
| Spine (Sag.) | 0.936 ( **0.055**) | 0.979 (-0.012) | 0.957 ( **0.024**) |
| 4CH | 0.943 ( **0.035**) | 0.970 ( 0.007) | 0.956 ( **0.022**) |
| 3VV | 0.694 ( **0.050**) | 0.722 (-0.014) | 0.708 ( **0.021**) |
| RVOT | 0.691 ( **0.029**) | 0.705 ( **0.044**) | 0.698 ( **0.036**) |
| LVOT | 0.925 ( **0.022**) | 0.933 ( **0.027**) | 0.929 ( **0.024**) |
| Background | 0.995 (-0.001) | 0.992 ( 0.007) | 0.993 ( 0.003) |

indeed learnt the most important feature for each class. We note, however, attention map outlines the discriminant region; in particular, it does not necessarily coincide with the entire object. This behaviour makes sense because some part of object will appear in background label (i.e. when the ideal plane is not reached). Qualitatively, however, the bounding boxes well agree with the annotated ground truth. Most crucially, the attention map is obtained for almost no additional computational cost; In comparison, [2] requires guided backpropagation for localisation, which limits the localisation speed. This highlights the advantage of attention model for the real-time applications.

## 4 Discussion

In this work, we considered soft-attention mechanism and discuss how to incorporate them into scan plane detection for fetal ultrasound to better exploit local structures. In particular, we highlighted several aspects: a normalisation strategy for the attention map, gridded attention mechanisms, and aggregation strategies. We empirically observed and reported that soft-max tends to generate a map that is over sensitive to local intensity changes, which is problematic as in medical imaging, image quality is often low. We found that dividing by sum helped attention to distribute more evenly. A Sigmoid functions is an alternative as it only normalises the range and allows more information to flow. However, we found that training is non-trivial due to the gradient saturation problem.

We noted that training the attention-mechanism was slightly more complex than the standard network architecture. In particular, we observed that the strategy employed to aggregate the attention maps at different scales affects both the learning of the attention mechanism itself and hence the performance.

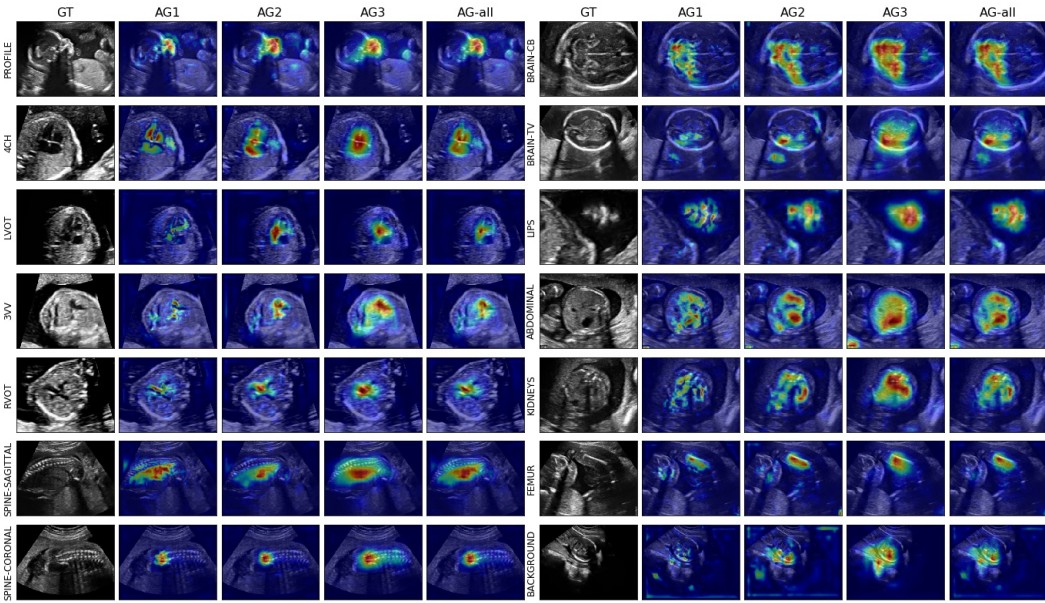

Figure 3: Examples of obtained attention map from AG-sononet. AG1 and AG2 are from layer 11 and 14 respectively. AG3 is obtained using CAM [33]. AG-all is obtained by normalising the maximum attented value across all AG's and taking mean over them.

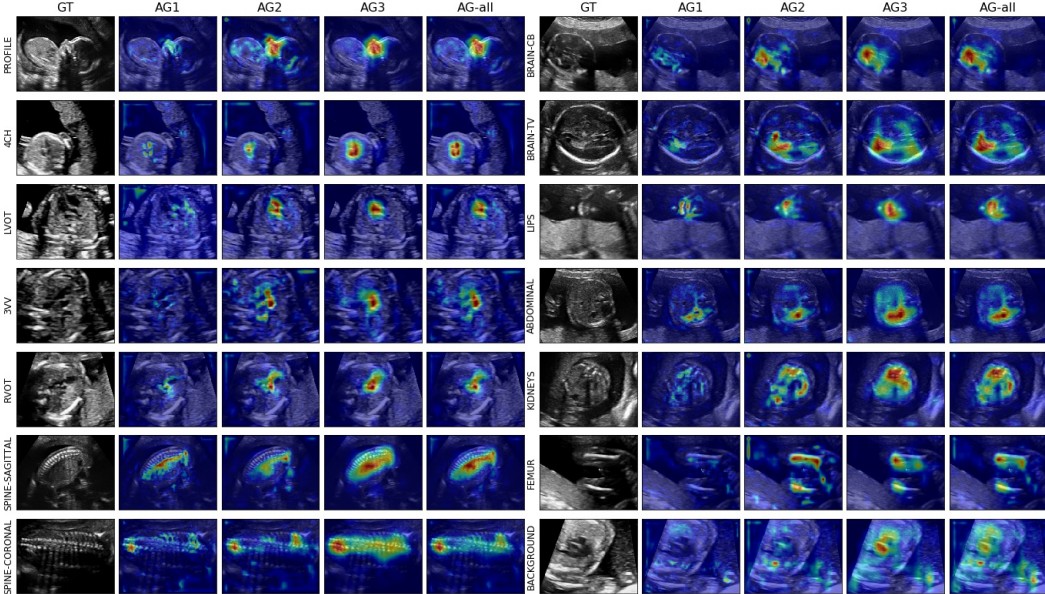

Figure 4: Examples of obtained attention map from AG-sononet-FT. AG1 and AG2 are from layer 11 and 14 respectively. AG3 is obtained using CAM [33]. AG-all is obtained by normalising the maximum attented value across all AG's and taking mean over them.

Having a loss term defined at each scale ensures that the network learns to attend at each scale. We observed that first training the network at each scale separately, followed by fine-tuning was the most stable approach to get the optimal performance.

The proposed network architecture resembles the one of deep-supervision in the sense that we add modules before the final layer which helps back-propagating the gradient at the early layers of the network. However, we argue that without a proper gating mechanism, we will not see any improvement. In fact, we saw that the model trained with deep supervision did not necessarily

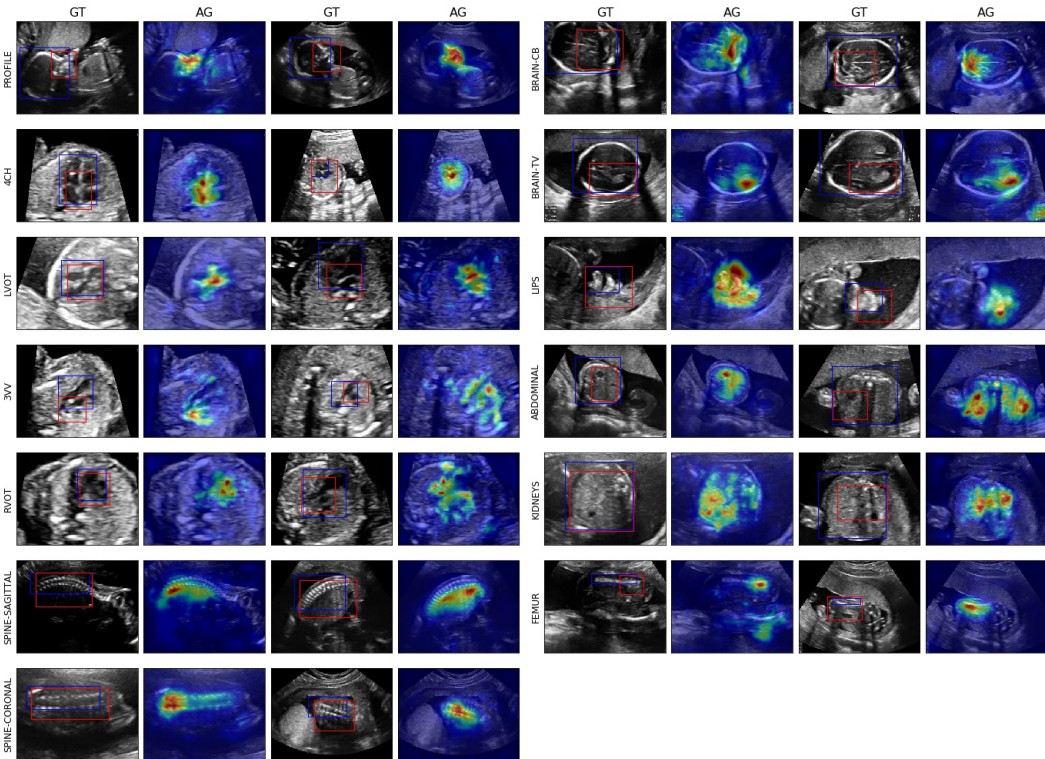

Figure 5: Examples of the obtained attention map and genereated bounding boxes (red) from AG-Sononet-FT across different subjects. The ground truth annotation is shown in blue. The detected region highly agrees with the object of interest.

obtain the best result. Therefore, while the network certainly benefits from backpropagating through additional pathways, the improvement in performance only came in conjunction with the attention mechanism.

## 5 Conclusion

In this work we proposed generalised self-attention mechanisms that can be easily incorporated into existing classification architectures. We applied the proposed architecture to standard scan plane detection during fetal ultrasound screening and showed that it improves overall results, especially precision, with much less parameters. This was done by generating the gating signal to pinpoint local as well as global information that is useful for the classification. Furthermore, it allows one to generate fine-grained attention map that can be exploited for object localisation. We envisage that the proposed soft-attention module will have great impact for explainable deep learning, weakly supervised object detection and/or segmentation – which are all vital research area for medical imaging analysis.

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

# 6 Appendix - Weakly Supervised Object Localisation (WSL)

In [2], WSL was performed by exploiting the pixel-level saliency map obtained by guilded-backpropagation, followed by ad-hoc procedure to extract bounding boxes. The same heuristics can be applied for the given network, however, owing to the attention map, we can device a much efficient way of performing object localisation. In particular, we generate object location by simply: (1) blur the attention maps, (2) threshold the low activations, (3) perform connected-component analysis, (4) select a component that overlaps at each scale and (5) apply bounding box around the selected components. In this heuristics, backpropagation is not required so it can be executed efficiently. We note, however, attention map outlines salient region used by the network to perform classification; in particular, it does not necessarily agree with the object of interest. This behaviour makes sense because some part of object will appear both in the class as well as background frame until the ideal plane is reached. Therefore, the quantitative result is shown in 3, however, the result is biased. We however define new metric called *Relative Correctness*, which is defined as 50% of maximum achievable IOU (due to bias). We see that in this metric, the method achieves very high results, indicating that it can detect relevant features of the object of interest in its proximity.

Table 3: WSL performance for the proposed strategy with AG-Sononet-FT-16. Correctness (Cor.) is defined as $IOU > 0.5$. Relative Correctness (Rel.) is defined as $IOU > 0.5 \times \max(IOU_{class})$.

|  | IOU Mean (Std) | Cor. (%) | Rel. (%) |
|---|---|---|---|
| Brain (Cb.) | 0.69 (0.11) | 0.96 | 0.96 |
| Brain (Tv.) | 0.68 (0.12) | 0.96 | 0.96 |
| Profile | 0.31 (0.08) | 0.00 | 0.80 |
| Lips | 0.42 (0.18) | 0.36 | 0.60 |
| Abdominal | 0.71 (0.10) | 0.96 | 0.96 |
| Kidneys | 0.73 (0.13) | 0.92 | 0.98 |
| Femur | 0.31 (0.11) | 0.02 | 0.58 |
| Spine (Cor.) | 0.53 (0.13) | 0.56 | 0.76 |
| Spine (Sag.) | 0.53 (0.11) | 0.54 | 0.94 |
| 4CH | 0.61 (0.14) | 0.76 | 0.86 |
| 3VV | 0.42 (0.14) | 0.34 | 0.62 |
| RVOT | 0.56 (0.15) | 0.70 | 0.76 |
| LVOT | 0.54 (0.15) | 0.62 | 0.80 |

