# OpenReview forum: "Attention-Gated Networks for Improving Ultrasound Scan Plane Detection"
_MIDL.amsterdam/2018/Conference — MIDL 2018 Poster_

### Review · AnonReviewer2 · 2018-05-01
**The authors propose a self-gated, soft-attention module, and integrate it into their previous network (SonoNet [2]) for real-time scan plane detection in fetal ultrasound screening.  Overall, the authors present an interesting and timely approach for end-to-end weakly-supervised object localization.**

**Rating:** 4
**Confidence:** 3

**Review:**

The attention mechanism introduced in this paper can yield fine-scale attention maps (or class activation maps) for efficient end-to-end object localization. However, the proposed method was not clearly described, and its effectiveness lacks of adequate experimental evaluation.

1)	The soft attention module proposed in this paper actually generalizes a similar approach in [6]. It is much easier to follow and figure out the difference/novelty if the authors could introduce the two parts separately.

2)	Also, to support their claims regarding the advantages of the generalized soft attention module (e.g., Eq. (2) and the specific normalization of attention coefficients), the authors needs to experimentally compare the performance with that of the original module [6].

3)	The authors claim that introducing W_f in (2) drives the fine-scale feature maps to focus more on learning discriminant features. This point lacks of justification/explanation.

4)	The evaluation & training of the proposed network should be described in more detail. For example, how was the partition of training/test data? Was it subject-specific or not? Also, how to select the background instances? Were “easy” and “hard” background instance (i.e., those contain anatomy of interests) equally treated?

5)   It should be interesting to experimentally check if the two attention branches could improve the discriminant capacity of the coarsest scale feature maps for the classification task.

6)	Figure 1 is misleading. It seems that the feature vectors after the attention layers should be produced via weighted average in term of attention coefficients, while not the global average pooling.


**Special Issue:**

Yes

---

### Review · AnonReviewer3 · 2018-05-09
**This paper presents a novel framework based on self-attention mechanisms for ultrasound scan plane detection. The proposed attention module is easily integrated into general pipeline and the attention map can be generated. The paper is well organized.**

**Rating:** 5
**Confidence:** 2

**Review:**


Quality & Clarity

#1. This paper is well organized, and methodology part as well as experiment section was clearly explained.
#2. The contribution of this paper is described very clearly.
#3. Experiment as well as visualization results are well organized and interesting.

Originality & Significance

(+) It is a technically interesting idea and the proposed module can be easily integrated into various framework.
(+) The proposed grid attention block are designed in a novel way and well incorporated to conventional classification algorithm.

**Special Issue:**

Yes

---

### Review · AnonReviewer1 · 2018-05-14
**An interesting and novel method with thorough descriptions and appropriate experimentation**

**Rating:** 4
**Confidence:** 2

**Review:**

I found the premise for this research to be well explained and relevant and the idea to be novel and interesting.  In general the paper is well written although some explanations and sentences are confusing and there are some typos and English language errors that should be checked.  (For example, the first attempt to summarise what is meant by "attention" -..... "Attention is a modular mechanism that allows to efficiently exploit localised information, which also provides soft object localisation during forward pass" is rather convoluted and could be clarified more carefully.)

I thought that the authors could have explained the data and the clinical problem a bit more carefully at the outset (what kind of data is acquired and what is the objective - i.e. the intended meaning of "scan plane detection").  Also in the description of the experimental data there are some oversights.  What precisely is labelled and how were the labels obtained?  Be explicit about how 2694 2D ultrasounds result in 122233+30553+38243 samples (how many samples per subject, why was it chosen to split this way?)

It is a disadvantage that the authors have a lot to describe within the space constraints.  In general section 3 is very text-heavy and could benefit from some bullet points or tables to make it more easily readable and understandable.  As a brief example The "Aggregation Strategy" section could be structured better by enumerating the 3 options (and naming conventions) within the text, even if space for bullet points cannot be found.

Improvement in the results is very small compared to the standard Sononet algorithm.  I would be interested to know whether the authors have checked for statistical significance.  I think the most interesting result is actually that the attention maps can be supplied with the authors method.

The contrast between figures 3 and 4 is difficult to make out.  It might be more useful to show the same images in each case with the different heat-maps to illustrate differences.

Could the authors describe any limitations or illustrate any cases of weak performance?

Some typos encountered (non-exhaustive):
Section 1.2, point 3 "....attention maps can BE used for fast...."
Section 3, results "In particular, AG-Sononets OBTAINS higher precision."
Section titled Attention Map Analysis and Object Localisation: "together WITH the bounding box annotation generated"

**Special Issue:**

No

---

### Comment · ~Bram_van_Ginneken1 · 2018-05-18
**Selection for longlist for special issue Medical Image Analysis**

Dear authors,

Congratulations on your acceptance to MIDL! We have selected your paper on the longlist for the Medical Image Analysis Special Issue. Please read this page:
https://midl.amsterdam/special-issue-in-medical-image-analysis/
Please answer the three questions that are listed on that page about your interest in submitting to the special issue, potential overlap with other publications, and related publications.

You can post your answer here directly below on openreview.net, or mail me directly at bram.vanginneken@radboudumc.nl.

Best regards, Bram

---

### Decision · Program_Chairs · 2018-05-15
**Paper102 Acceptance Decision**

Poster